

# Power1D: a Python toolbox for numerical power estimates in experiments involving one-dimensional continua

Todd C. Pataky

Institute for Fiber Engineering, Shinshu University, Ueda, Japan

## ABSTRACT

The unit of experimental measurement in a variety of scientific applications is the one-dimensional (1D) continuum: a dependent variable whose value is measured repeatedly, often at regular intervals, in time or space. A variety of software packages exist for computing continuum-level descriptive statistics and also for conducting continuum-level hypothesis testing, but very few offer power computing capabilities, where 'power' is the probability that an experiment will detect a true continuum signal given experimental noise. Moreover, no software package yet exists for arbitrary continuum-level signal/noise modeling. This paper describes a package called **power1d** which implements (a) two analytical 1D power solutions based on random field theory (RFT) and (b) a high-level framework for computational power analysis using arbitrary continuum-level signal/noise modeling. First **power1d**'s two RFT-based analytical solutions are numerically validated using its random continuum generators. Second arbitrary signal/noise modeling is demonstrated to show how **power1d** can be used for flexible modeling well beyond the assumptions of RFT-based analytical solutions. Its computational demands are non-excessive, requiring on the order of only 30 s to execute on standard desktop computers, but with approximate solutions available much more rapidly. Its broad signal/noise modeling capabilities along with relatively rapid computations imply that **power1d** may be a useful tool for guiding experimentation involving multiple measurements of similar 1D continua, and in particular to ensure that an adequate number of measurements is made to detect assumed continuum signals.

## INTRODUCTION

Analyzing multiple measurements of one-dimensional (1D) continua is common to a variety of scientific applications ranging from annual temperature fluctuations in climatology (Fig. 1) to position trajectories in robotics. These measurements can be denoted $y(q)$ where $y$ is the dependent variable, $q$ specifies continuum position, usually in space or time, and where the continua are sampled at $Q$ discrete points. For the climate data depicted in Fig. 1 $y$ is temperature, $q$ is day and $Q = 365$.

Measurements of $y(q)$ are often: (i) registered and (ii) smooth. The data are 'registered' in the sense that point $q$ is homologous across multiple continuum measurements.

Corresponding author
Todd C. Pataky,
tpataky@shinshu-u.ac.jp

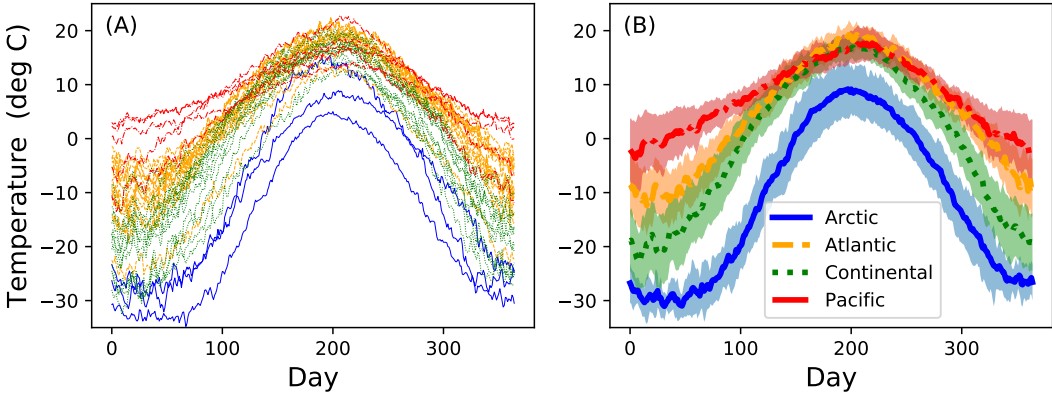

**Figure 1   Canadian temperature data (*Ramsay & Silverman, 2005*).** (A) All measurements. (B) Means (thick lines) and standard deviations (error clouds). Dataset download on 28 March 2017 from: http://www.psych.mcgill.ca/misc/fda/downloads/FDAfuns/Matlab/fdaMatlab.zip (./examples/weather)

Registration implies that it is generally valid to compute mean and variance continua as estimators of central tendency and dispersion (*Friston et al., 1994*). That is, at each point $q$ the mean and variance values are computed, and these form mean and variance continua (Fig. 1B) which may be considered unbiased estimators of the true population mean and variance continua.

The data are 'smooth' in the sense that continuum measurements usually exhibit low frequency signal. This is often a physical consequence of the spatial or temporal process which $y(q)$ represents. For example, the Earth's rotation is slow enough that day-to-day temperature changes are typically much smaller than season-to-season changes (Fig. 1A). Regardless of the physical principles underlying the smoothness, basic information theory in fact *requires* smooth continua because sufficiently high measurement frequency is needed to avoid signal aliasing. This smoothness has important statistical implications because smoothness means that neighboring points ($q$ and $q+1$) are correlated, or equivalently that adjacent points do not vary in a completely independent way. Thus, even when $Q$ separate values are measured to characterize a single continuum, there may be far fewer than $Q$ independent stochastic units underlying that continuum process.

The Canadian temperature dataset in Fig. 1 exhibits both features. The data are naturally registered because each measurement station has one measurement per day over $Q = 365$ days. The data are smooth because, despite relatively high-frequency day-to-day temperature changes, there are also comparatively low-frequency changes over the full year and those low-frequency changes are presumably the signals of interest.

Having computed mean and variance continua it is natural to ask probabilistic questions regarding them, and two basic kinds of probability questions belong to the categories: (i) classical hypothesis testing and (ii) power analysis. Continuum-level hypothesis testing has been well-documented in the literature (*Friston et al., 1994*; *Nichols & Holmes, 2002*; *Pataky, 2016*) but power has received comparatively less attention. While this paper focuses on power analysis it is instructive to first consider continuum-level hypothesis testing because those results are what power analysis attempts to control.

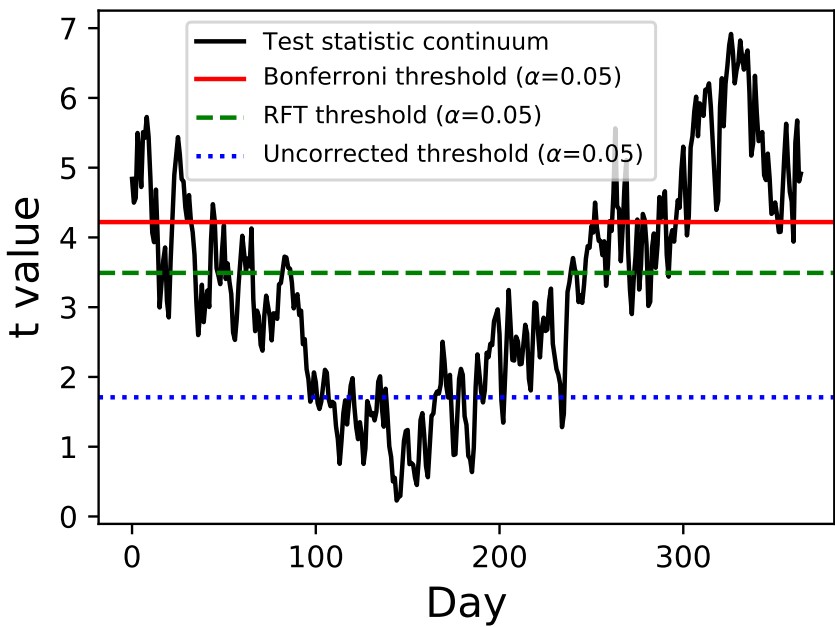

**Figure 2** Two-sample hypothesis test comparing the Atlantic and Continental regions from **Fig. 1**. The test statistic continuum is depicted along with uncorrected, random field theory (RFT)-corrected and Bonferroni-corrected thresholds.

## Continuum-level hypothesis testing

Classical hypothesis testing can be conducted at the continuum level using a variety of theoretical and computational procedures. In the context of the temperature data (Fig. 1B) a natural hypothesis testing question is: *is there is a statistically significant difference between the Atlantic and Continental mean temperature continua?* Answering that question requires a theoretical or computational model of stochastic continuum behavior so that probabilities pertaining to particular continuum differences can be calculated.

One approach is Functional Data Analysis (FDA) (*Ramsay & Silverman, 2005*) which combines 'basis functions', or mathematically-defined continua, to model the data. Since the basis functions are analytical, one can compute a variety of probabilities associated with their long-term stochastic behavior. A second approach is Random Field Theory (RFT) (*Adler & Hasofer, 1976*; *Hasofer, 1978*) which extends Gaussian behavior to the 1D continuum level via a smoothness parameter (*Kiebel et al., 1999*) from which a variety of continuum level probabilities can be calculated (*Friston et al., 1994*). A third approach is the non-parametric permutation method of *Nichols & Holmes (2002)* which, instead of modeling stochastic continuum behavior directly, instead constructs probability distributions through iterative computation. Ultimately these and all other approaches, when used for classical hypothesis testing, offer a correction for multiple comparisons across the $Q$ continuum nodes based on continuum smoothness.

Example hypothesis testing results for the Canadian temperature data are depicted in Fig. 2. Since there are mean and variance continua it is trivial to compute the test statistic continuum, here as the two-sample t statistic representing the variance-normalized

difference between the Atlantic and Continental regions. The next step is less trivial: finding the critical test statistic threshold. The threshold is meant to represent the value above which purely random test statistic continua (i.e., those produced by random continua when the true continuum difference is null) would traverse in $\alpha$ percent of an infinite number of experiments, where $\alpha$ is the Type I error rate and is usually 0.05.

Of the three thresholds depicted in Fig. 2 only one (the RFT threshold) is a true continuum-level threshold. The other two depict nappropriate thresholds as references to highlight the meaning of the RFT threshold. In particular, the uncorrected threshold ($\alpha = 0.05$) is 'uncorrected' because it presumes $Q = 1$; since $Q = 365$ for these data it is clearly inappropriate. On the other extreme is a Bonferroni threshold which assumes that there are $Q$ completely independent processes. It is a 'corrected' threshold because it acknowledges that $Q > 1$, but it is inappropriate because it fails to account for continuum smoothness, and thus overestimates the true number of stochastic processes underlying these data. The third method (RFT) is also a 'corrected' threshold, and it is closest to the true threshold required to control $\alpha$ because it considers both $Q$ and smoothness *Friston et al. (1994)*. Specifically, it assesses inter-node correlation using the 1D derivative (*Kiebel et al., 1999*) to lower the estimated number of independent processes, which in turn lowers the critical threshold relative to the Bonferroni threshold. This RFT approach is described extensively elsewhere *Friston et al. (2007)* and has also been validated extensively for 1D continua (*Pataky, 2016*).

For this particular dataset the test statistic continuum crosses all three thresholds, implying that the null hypothesis of equivalent mean continua is rejected regardless of correction procedure. If the continuum differences are not as pronounced as they are here, especially near the start and end of the calendar year, the correction procedure would become more relevant to interpretation objectivity.

## Continuum-level power analysis

Before conducting an experiment for which one intends to conduct classical hypothesis testing it is often useful to conduct power analysis, where 'power' represents the probability of detecting a true effect. The main purposes of power analysis are (a) to ensure that an adequate number of measurements is made to elucidate a signal of empirical interest and (b) to ensure that not too many measurements are made, in which case one risks detecting signals that are not of empirical interest. The balance point between (a) and (b) is conventionally set at a power of 0.8, and that convention is followed below.

The literature describes two main analytical approaches to continuum-level power analysis: (i) inflated variance (*Friston et al., 1996*) and (ii) noncentral RFT (*Hayasaka et al., 2007*; *Mumford & Nichols, 2008*; *Joyce & Hayasaka, 2012*). The inflated variance method models signal as smooth Gaussian noise (Fig. 3A) which is superimposed upon Gaussian noise with different amplitude and smoothness. The non-central RFT approach models signal as a constant mean shift from the null continuum (Fig. 3B). Since both techniques are analytical power calculations can be made effectively instantaneously. However, both techniques are limited by simple signal models and relatively simple noise models. In reality

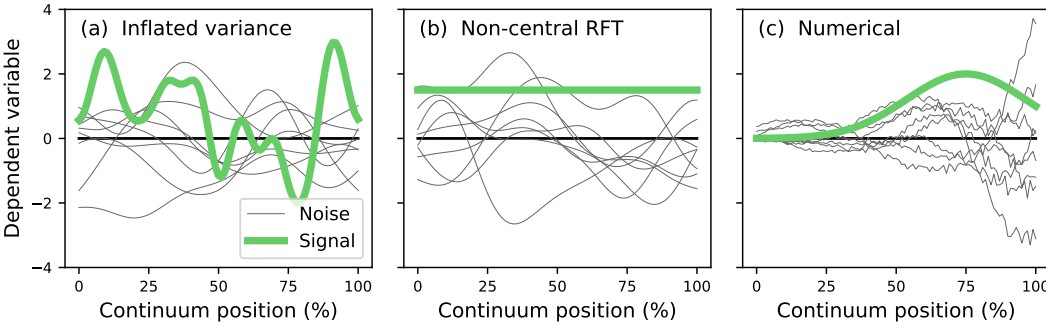

**Figure 3** **Continuum-level power analysis methods.** (A) *Friston et al. (1996)*. (B) *Hayasaka et al. (2007)* and *Mumford & Nichols (2008)*. (C) This paper's proposed computational method. RFT = random field theory.

the signal can be geometrically arbitrary and the noise can be arbitrarily complex (Fig. 3C). Currently no analytical methods exist for arbitrary signal geometries and arbitrary noise.

The purpose of this study was to develop a computational approach to continuum-level power analysis that permits arbitrary signal and noise modeling. This paper introduces the resulting open-source Python software package called **power1d**, describes its core computational components, and cross-validates its ultimate power results with results from the two existing analytical methods (inflated variance and non-central RFT). Source code, HTML documentation and scripts replicating all results in this manuscript are available at http://www.spm1d.org/power1d.

## SOFTWARE IMPLEMENTATION

**power1d** was developed in Python 3.6 (*Van Rossum, 2014*) using Anaconda 4.4 (*Continuum Analytics, 2017*) and is also compatible with Python 2.7. Its dependencies include Python's standard numerical, scientific and plotting packages:

- NumPy 1.11 (*Van der Walt, Colbert & Varoquaux, 2011*).
- SciPy 0.19 (*Jones, Oliphant & Peterson, 2001*).
- matplotlib 2.0 (*Hunter, 2007*).

Other versions of these dependencies are likely compatible but have not been tested thoroughly. The package is organized into the following modules:

- **power1d**.*geom*—1D geometric primitives for data modeling.
- **power1d**.*models*—high-level interfaces to experiment modeling and numerical simulation.
- **power1d**.*noise*—1D noise classes including mixtures, signal-dependent and compound classes.
- **power1d**.*prob*—analytical probabilities for central and noncentral t and F fields.
- **power1d**.*random*—smooth 1D Gaussian field generation.
- **power1d**.*roi*—regions-of-interest (ROIs) for geometric hypothesis constraints.
- **power1d**.*stats*—standard t and F computations for continua.

Details regarding the contents and capabilities of each module are provided in **power1d**'s documentation (http://www.spm1d.org/power1d) and are summarized below, building to a model and ultimately a power analysis of the Canadian temperature dataset above (Fig. 1).

### Geometry (power1d.geom)

Basic geometries can be constructed and visualized as follows:

```
import power1d

Q = 101
y = power1d.geom.GaussianPulse( Q , q=60 , fwhm=20, amp=3.2 )
y.plot()
```

Here $Q$ is the continuum size, $q$ is the continuum position at which the Gaussian pulse is centered, *fwhm* is the full-width-at-half-maximum of the Gaussian kernel, and *amp* is its maximum value (Fig. 4). All of **power1d**'s geometric primitives have a similar interface and are depicted in Fig. 5. More complex geometries can be constructed using standard Python operators as follows (see Fig. 6).

```
import power1d

Q  = 101
y0 = power1d.geom.GaussianPulse( Q , q=40 , fwhm=60, amp= 1 )
y1 = power1d.geom.Sinusoid( Q , amp=1 , hz=2 )

yA = y0 + y1
yB = y0 * y1
yC = y0 ** y1
```

### Noise (power1d.noise)

Continuum-level noise objects can be constructed and visualized as follows:

```
from matplotlib import pyplot
import power1d

J = 8
Q = 101
n0 = power1d.noise.Gaussian( J , Q , mu=0 , sigma=1 )
n1 = power1d.noise.SmoothGaussian( J , Q , mu=0 , sigma=1 , fwhm=20 )

ax0 = pyplot.subplot( 121 )
ax1 = pyplot.subplot( 122 )
n0.plot( ax=ax0 )
n1.plot( ax=ax1 )
```

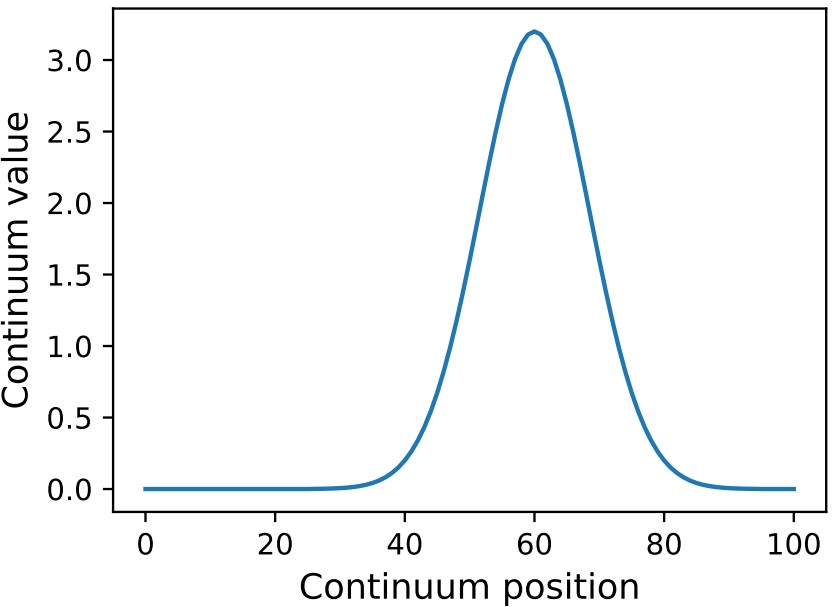

**Figure 4** Example GaussianPulse geometry.

Here *J* is sample size and is a necessary input for all power1d.noise classes. This code chunk results in the noise depicted in Fig. 7. The SmoothGaussian noise (Fig. 7B) represents residuals observed in real datasets like those depicted implicitly in Fig. 1A. For this SmoothGaussian noise model the *fwhm* parameter represents the full-width-at-half-maximum of a Gaussian kernel that is convolved with uncorrelated Gaussian continua. RFT describes probabilities associated with smooth Gaussian continua (Fig. 7B) and in particular the survival functions for test statistic continua (*Friston et al., 1996*; *Pataky, 2016*).

All **power1d** noise models are depicted in Fig. 8. Compound noise types are supported including additive, mixture, scaled and signal-dependent. As an example, the additive noise model depicted in Fig. 8H can be constructed as follows:

```
n0 = power1d.noise.Gaussian( J , Q , mu=0 , sigma=0.1 )
n1 = power1d.noise.SmoothGaussian( J , Q , mu=0 , sigma=1.5 , fwhm=40 )
n  = power1d.noise.Additive ( noise0 , noise1 )
```

All noise models use the *random* method to generate new random continua, and all store the current continuum noise in the *value* attribute, and all number generation can be controlled using NumPy's *random.seed* method as follows:

```
np.random.seed(0)
J = 10
Q = 101

noise = power1d.noise.Gaussian ( J , Q , mu=0 , sigma=1 )
print( noise.value[ 0 , 0:3 ] )
```

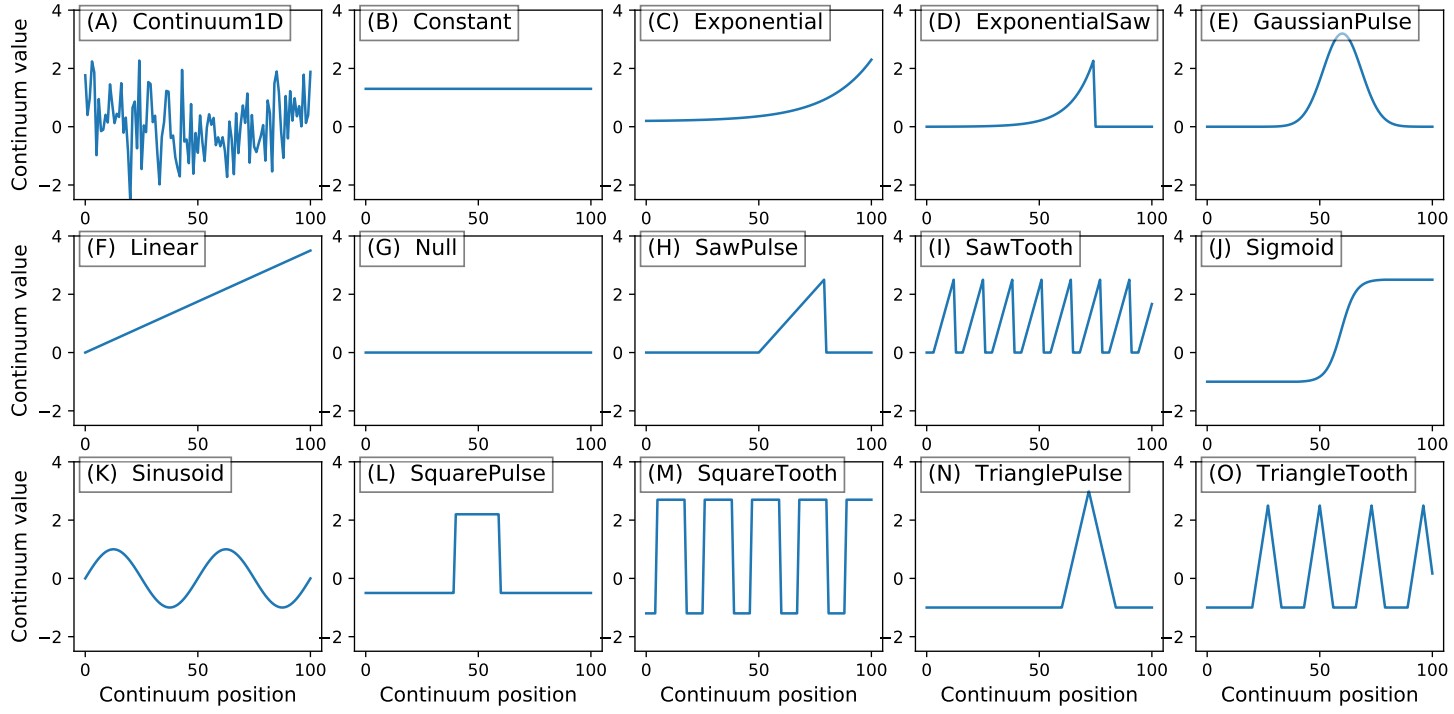

**Figure 5 All geometric primitives.** The Continuum1D primitive accepts an arbitrary 1D array as input, and all other primitives are parameterized. (A) Continuum1D, (B) Constant, (C) Exponential, (D) ExponentialSaw, (E) GaussianPulse, (F) Linear, (G) Null, (H) SawPulse, (I) SawTooth, (J) Sigmoid, (K) Sinusoid, (L) SquarePulse, (M) SquareTooth, (N) TrianglePulse, (O) TriangleTooth.

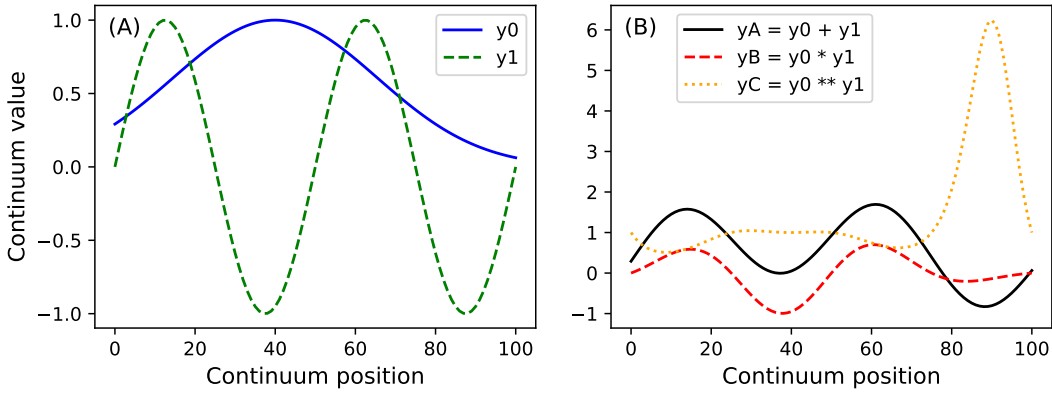

**Figure 6 (A)** Two example geometric primitives. **(B)** Python operators used to construct complex geometries from primitives.

```
noise.random()
print( noise.value[ 0 , 0:3 ] )

np.random.seed(0)
noise.random( )
print( noise.value [ 0 , 0:3 ] )
```

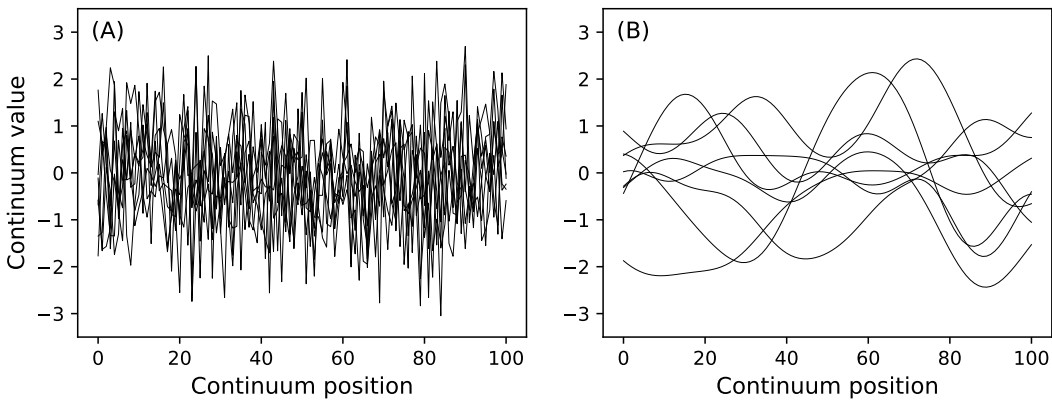

**Figure 7** (A) Uncorrelated Gaussian noise. (B) Smooth (correlated) Gaussian noise.

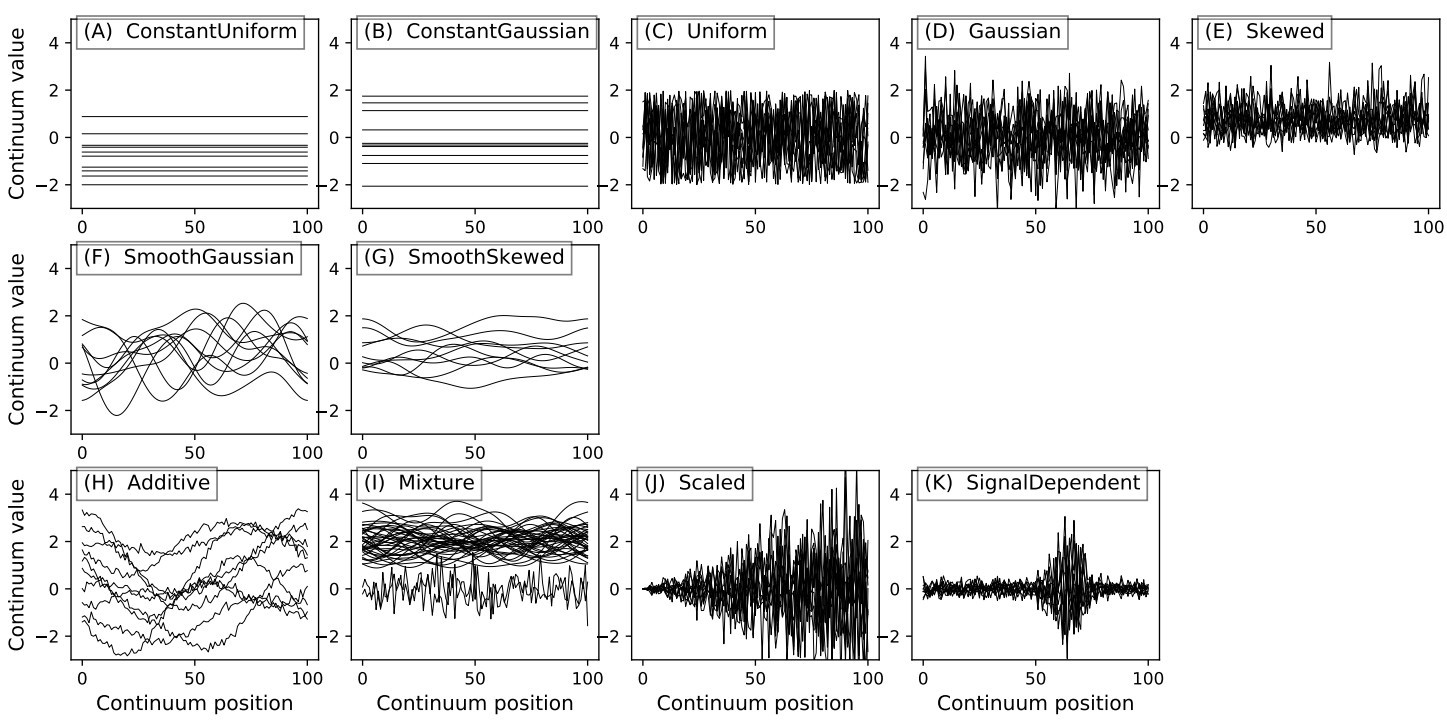

**Figure 8** All noise models. (A–E), (F–G) and (H–K) depict basic, smooth and compound noise types, respectively.

The first, second and third *print* commands display the following results:

```
[  0.176   0.040   0.098 ]
[ -0.100   0.167   0.016 ]
[  0.176   0.040   0.098 ]
```

This emphasizes control over **power1d**'s random values via *np.random.seed*.

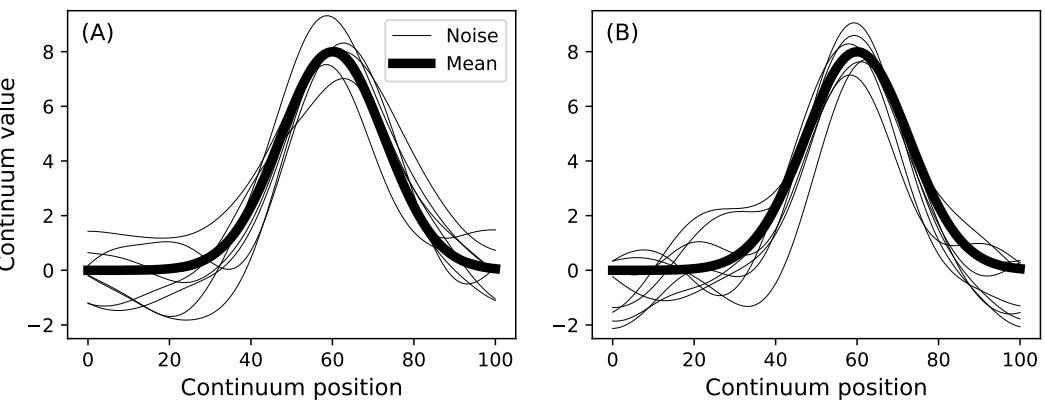

**Figure 9** **Data sample model.** (A) and (B) depict two separate samples generated using a single DataSample object.

## Data sample and experiment models (power1d.models)

In this section the terms "DataSample" and "data sample" refer to the object class: **power1d.models.DataSample** and a numerical instantiation of that class, respectively. DataSample objects have three components: (a) baseline, (b) signal and (c) noise. The first two are power1d.geom objects and the last is a power1d.noise object. DataSample objects, like noise objects, use *random* to generate new random data samples as follows:

```
J = 8
Q = 101

baseline = power1d.geom.Null( Q )
signal = power1d.geom.GaussianPulse( Q , q=60 , fwhm=30, amp=8 )
noise = power1d.noise.SmoothGaussian( J , Q , mu=0 , sigma=1 , fwhm=20 )

model = power1d.models.DataSample( baseline , signal , noise )
model.plot()

model.random()
model.plot()
```

Two such data samples constructed in this manner are depicted in Fig. 9. Any geometry object and any noise object can be combined to form a DataSample object. The purpose of the *baseline* object is to provide a visual reference when constructing DataSample models. For example, the Atlantic temperature data from Fig. 1A could be modeled with the experimentally observed mean as follows:

```
data = power1d.data.weather()
y = data[ " Continental " ]

J = 8
```

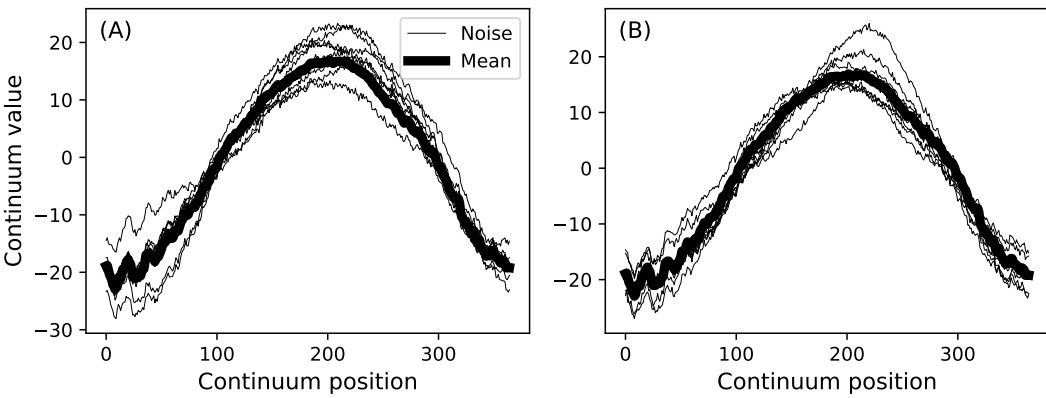

**Figure 10  Data sample model using experiment mean from the Continental data in Fig. 1.** (A) and (B) depict two separate randomly generated samples.

```
Q = 365

baseline = power1d.geom.Continuum1D( y.mean( axis=0 ) )
signal = power1d.geom.Null( Q )

n0 = power1d.noise.Gaussian( J , Q , mu=0 , sigma=0.3 )
n1 = power1d.noise.SmoothGaussian( J , Q , mu=0 , sigma=3 , fwhm=70 )
noise = power1d.noise.Additive( n0 , n1 )

model=power1d.models.DataSample( baseline , signal , noise , J=J )
model.plot()

model.random()
model.plot()
```

The first command loads the Canadian temperature (or 'weather') data as a Python dictionary. The second extracts just the Continental data. Next the experimental mean is used to create a *Continuum1D* baseline object, and a *Null* signal object is also created. The next three lines create an additive noise model which contains both high- and low-frequencies. Subsequently a *DataSample* model is created with the sample size *J*. The results of this code chunk are depicted in Fig. 10.

The *baseline* component of DataSample objects have no effect on subsequently described power calculations, which are based on the *signal* and *noise* components. The *baseline* component is included for two reasons: (a) to visually guide DataSample construction, and (b) to permit hypothesis-relevant calculations. For example, one's hypothesis may pertain to a function of the continua like their cumulative integral rather than to the originally measured continua themselves. In that case the *baseline*'s magnitude as well as its positive and negative regions could be important for test statistic computation.

Once a data sample model is constructed it can be routed into an Experiment model for simulation as follows:

```
J = 8
Q = 101

baseline = power1d.geom.Null( Q )
signal = power1d.geom.GaussianPulse( Q , q=60 , fwhm=30, amp=3 )
noise = power1d.noise.SmoothGaussian( J , Q , mu=0 , sigma=1 , fwhm=20 )

model = power1d.models.DataSample( baseline , signal , noise , J=J )

teststat = power1d.stats.t_1sample
emodel = power1d.models.Experiment( model , teststat )
emodel.simulate( 50 )

pyplot.plot( emodel.Z.T , color="k" , linewidth=0.5 )
```

Here *teststat* is a function that computes the one-sample *t* statistic continuum. The *Experiment* model contains both a *DataSample* model and a test statistic computer. Once the *simulate* method is called, a random data sample is generated and the corresponding test statistic continuum is calculated and stored in the *Z* attribute, in this case for a total of 50 iterations.

The resulting test statistic continua are depicted in Fig. 11. Since test statistic continua can be numerically generated in this manner for arbitrary *DataSample* and *Experiment* models, it follows that power analysis can be numerically conducted by comparing two experiment models, one representing the null hypothesis (which contains null signal) and one representing the alternative hypothesis (which contains the signal one wishes to detect). **power1d** provides a high-level interface to that two-experiment comparison through its *ExperimentSimulator* object as demonstrated below:

```
J = 5
Q = 101

baseline = power1d.geom.Null( Q )
signal0 = power1d.geom.Null( Q )
signal1 = power1d.geom.GaussianPulse( Q , q=60 , fwhm=30, amp=2 )
noise = power1d.noise.Gaussian( J , Q , mu=0 , sigma=1 )

model0 = power1d.models.DataSample( baseline , signal0 , noise , J=J )
model1 = power1d.models.DataSample( baseline , signal1 , noise , J=J )

teststat = power1d.stats.t_1sample
emodel0 = power1d.models.Experiment( model0 , teststat )
```

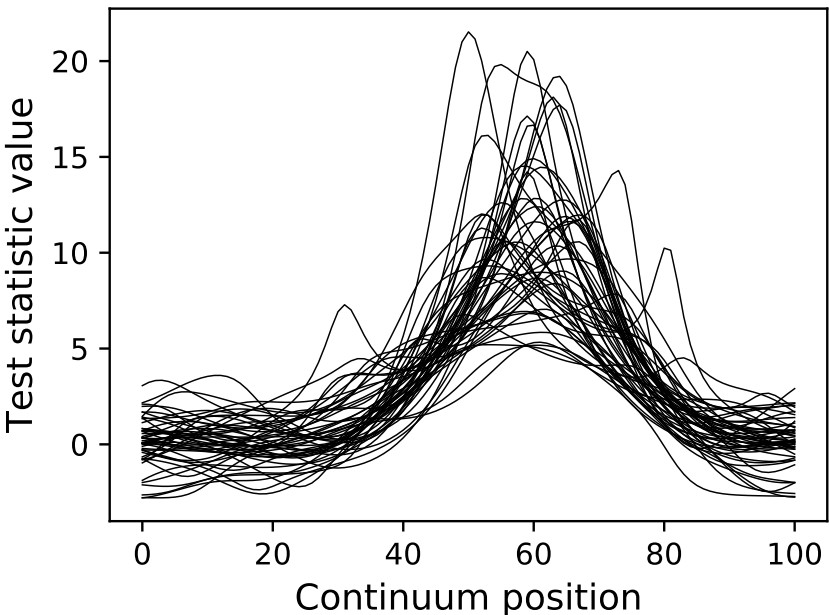

**Figure 11** Test statistic continua generated using an Experiment model (50 iterations).

```
emodel1 = power1d.models.Experiment( model1 , teststat )

sim = power1d.ExperimentSimulator( emodel0 , emodel1 )
results = sim.simulate( 10000 )
results.plot()
```

Note that the *emodel0* and *emodel1* objects represent null and a Gaussian pulse signal, respectively, and thus represent the null and alternative hypotheses, respectively. The Monte Carlo simulation proceeds over 10,000 iterations (triggered by the *simulate* command) and completes for this example in approximately 2.2 s. The final *results.plot* command produces the results depicted in Fig. 12.

In this example the omnibus power is 0.92 (Fig. 12A), implying that the probability of rejecting the null at at least one continuum location is 0.92. This omnibus power should be used when the hypothesis pertains to the entire continuum because it embodies whole-continuum-level control of both false negatives and false positives.

While the omnibus power is greater than 0.9, the point-of-interest (POI) and center-of-interest (COI) powers are both well below 0.8 (Fig. 12C); see the Fig. 12 caption for a description of POI and COI powers. The POI power should be used if one's hypothesis pertains to a single continuum location. The COI power should be used if the scope of the hypothesis is larger than a single point but smaller than the whole continuum.

Overall these results imply that, while the null hypothesis will be rejected with high power, it will not always be rejected in the continuum region which contains the modeled signal (i.e., roughly between continuum positions 40 and 80). This simple model thus highlights the following continuum-level power concepts:

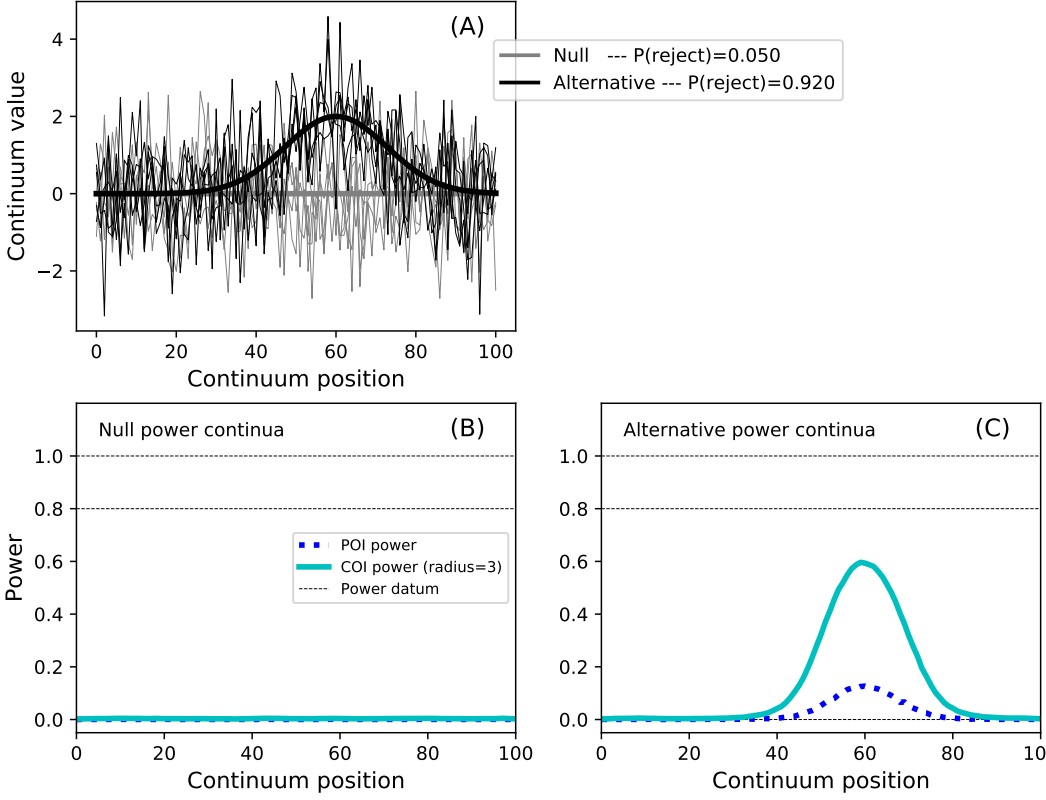

**Figure 12 Example power1d results ($\alpha = 0.05$).** (A) depicts the two experiment models and the omnibus power ($p = 0.920$). (B, C) depict power continua (B: null model, C: alternative model). The point-of-interest (POI) continuum indicates the probability of null hypothesis rejection at each continuum point. The center-of-interest (COI) continuum depicts the same but expands the search area to a certain radius surrounding the POI, in this case with an arbitrary radius of three. Thus the omnibus power is equivalent to the maximum COI power when the COI radius is $Q$ (i.e., the full continuum size). The integral of the POI power continuum for the null model is $\alpha$. Powers of 0, 0.8 and 1 are displayed as dotted lines for visual reference.

- Continuum-level signals can be modeled with arbitrary geometry.
- Continuum-level omnibus power does not necessarily pertain to the modeled signal.
- The investigator must specify the scope of the hypothesis in an *a priori* manner (i.e., single point, general region or whole-continuum) and use the appropriate power value (i.e., POI, COI or omnibus, respectively).

The model depicted in Fig. 12 is simple, and similar results could be obtained analytically by constraining the continuum extent of noncentral RFT inferences (*Hayasaka et al., 2007*). The advantages of numerical simulation are thus primarily for situations involving arbitrary complexities including but not limited to: multiple, possibly interacting signals, signal-dependent noise, covariate-dependent noise, unequal sample sizes, non-sphericity, etc. All of these complexities introduce analytical difficulties, but all are easily handled within **power1d**'s numerical framework.

### Regions of interest (power1d.roi)

The final functionality supported in **power1d** is hypothesis constraining via region of interest (ROI) continua. In practical applications, even when complete continua are recorded, one's hypothesis does not necessarily relate to the whole continuum. For example, the Canadian temperature example (Fig. 1) depict daily values collected for the whole year, but one's hypothesis might pertain only to the summer months (approximately days 150–250). In this case it is probably most practical to model the entire year, but constrain the hypothesis to a certain portion of it as follows:

```
data = power1d.data.weather()
y = data[ " Continental " ]

baseline = power1d.geom.Continuum1D( y.mean ( axis=0 ) )
signal0 = power1d.geom.Null( Q )
signal1 = power1d.geom.GaussianPulse( Q , q=200 , amp=6 , fwhm=100 )

n0 = power1d.noise.Gaussian( J , Q , mu=0 , sigma=0.3 )
n1 = power1d.noise.SmoothGaussian( J , Q , mu=0 , sigma=5 , fwhm=70 )
noise = power1d.noise.Additive( n0 , n1 )

model0 = power1d.models.DataSample( baseline , signal0 , noise , J=J )
model1 = power1d.models.DataSample( baseline , signal1 , noise , J=J )
teststat = power1d.stats.t_1sample
emodel0 = power1d.models.Experiment( model0 , teststat )
emodel1 = power1d.models.Experiment( model1 , teststat )

sim = power1d.ExperimentSimulator( emodel0 , emodel1 )
results = sim.simulate( 10000 )

roi = np.array( [ False ] * Q )
roi[ 150 : 250 ] = True

results.set_roi( roi )
results.set_coi_radius( 50 )
results.plot()
```

The code above models a maximum temperature increase of six degrees on Day 200 as a Gaussian pulse with an FWHM of 100 days, and constrains the hypothesis to Days 150–250 via the *set_roi* method. The results in Fig. 13 depict the ROI as blue background window and suggest that the omnibus power is close to 0.8. Setting the COI radius to the ROI radius of 50 via the *set_coi_radius* method emphasizes that the COI power continuum's maximum is the same as the omnibus power. Also note that, had an ROI not been set, the ROI is implicitly the entire continuum, in which case the omnibus power would have been

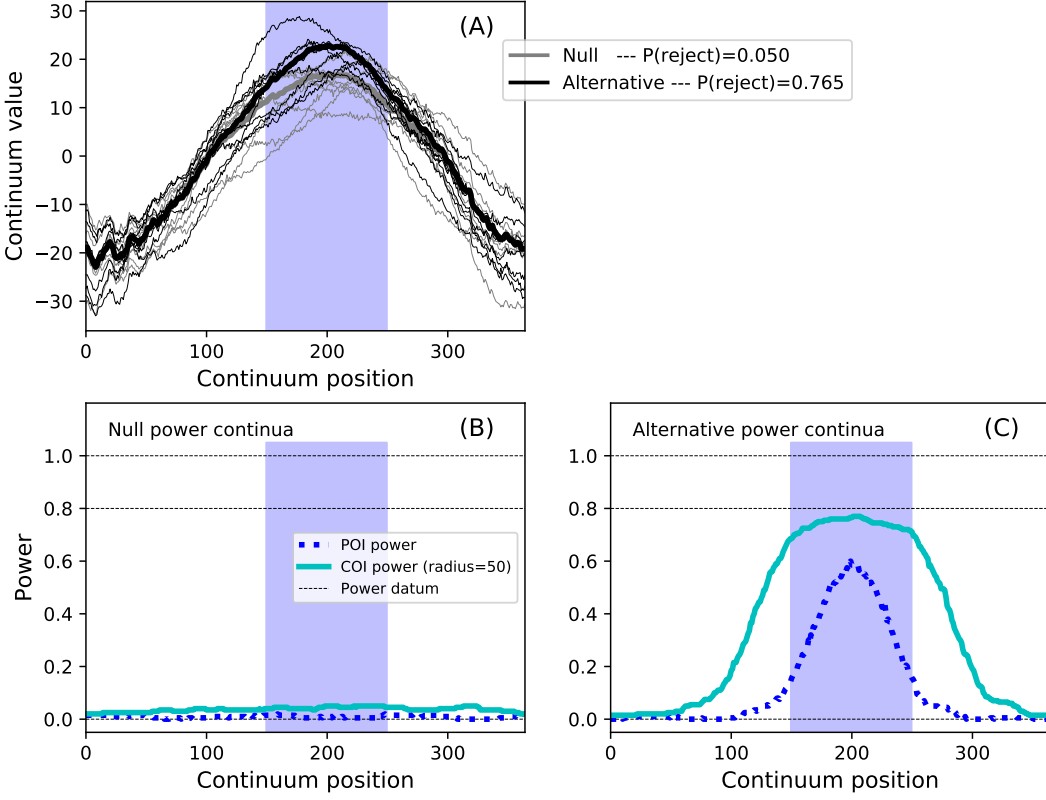

**Figure 13** **Example region of interest (ROI)-constrained power results ($\alpha = 0.05$).** Note that an COI radius of 365 would raise the null COI power continuum to $\alpha$. (A) depicts the two experiment models and the omnibus power ($p = 0.765$). (B, C) depict power continua (B: null model, C: alternative model).

considerably lower at 0.586. This emphasizes the fact that the critical threshold must be raised as the continuum gets larger in order to control for omnibus false positives across the continuum. These analyses, involving a more complex additive noise model and 10,000 iterations, required approximately 15 s on a standard desktop PC.

## VALIDATIONS

### 0D power

**power1d** can be used for standard 0D (scalar) power assessments by setting an ROI object to a single continuum point as follows. First set values for the main power-relevant parameters:

```
alpha = 0.05
J = 12
effect = 0.8
df = J − 1
delta = effect * J ** 0.5
```

where *alpha*, *df* and *delta* are the Type I error rate, degrees of freedom and noncentrality parameter, respectively. Next compute power analytically:

```
from scipy import stats
u = stats.t.isf( alpha , df )
p = stats.nct.sf( u , df , delta )
```

where $u$ is the critical threshold and where the power is $p = 0.829$. To replicate this in **power1d** one must create a model which replicates the assumptions underlying the analytical calculation above. In the code below a continuum size of $Q = 2$ is used because that is the minimum size that **power1d** supports.

```
Q = 2
baseline = power1d.geom.Null( Q )
signal0 = power1d.geom.Null( Q )
signal1 = power1d.geom.Constant( Q , amp=effect )
noise = power1d.noise.Gaussian( J , Q , mu=0 , sigma=1 )
model0 = power1d.DataSample( baseline , signal0 , noise , J=J )
model1 = power1d.DataSample( baseline , signal1 , noise , J=J )
```

Last, simulate the modeled experiments and numerically estimate power:

```
teststat = power1d.stats.t_1sample
emodel0 = power1d.models.Experiment( model0 , teststat )
emodel1 = power1d.models.Experiment( model1 , teststat )
sim = power1d.ExperimentSimulator( emodel0 , emodel1 )
results = sim.simulate( 1000 )
roi = np.array( [ True , False ] )
results.set_roi( roi )
p = results.p_reject1
```

Here power is given by the *p_reject1* attribute of the simulation results (i.e., the probability of rejecting the null hypothesis in alternative experiment given the null and alternative models) and in this case the power is estimated as $p = 0.835$. Increasing the number of simulation iterations improves convergence to the analytical solution.

Repeating across a range of sample and effect sizes yields the results depicted in Fig. 14. This **power1d** interface for computing 0D power is admittedly verbose. Nevertheless, as a positive point **power1d**'s interface emphasizes the assumptions that underly power computations, and in particular the nature of the signal and noise models.

### 1D power: inflated variance method

The inflated variance method (*Friston et al., 1996*) models signal as a Gaussian continuum with a particular smoothness and particular variance. **power1d** does not support random signal modeling, but the inflated variance model can nevertheless be modeled using alternative noise models as demonstrated below. First all power-relevant parameters are set:

```
J = 20
```

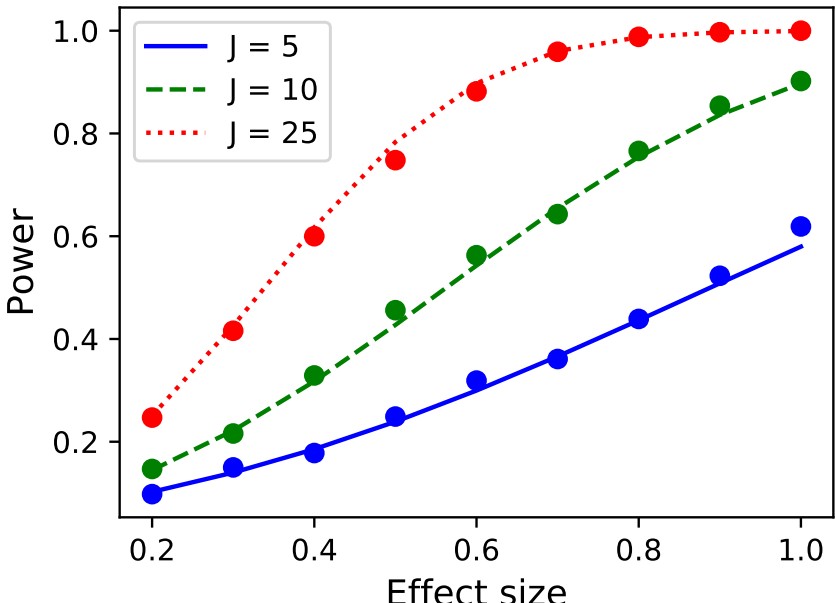

**Figure 14** **Validation of power1d's 0D power calculations.** Solid lines depict theoretical solutions from the noncentral t distribution and dots depict **power1d**'s numerically simulated results (1,000 iterations each).

```
Q = 201
df = J - 1
alpha = 0.05
W0 = 20
W1 = 10.0
sigma = 2.0
```

Here *W0* and *W1* are the continuum smoothness values under the null and alternative hypotheses, respectively, and *sigma* is the effect size as the standard deviation of the 'signal' (i.e., noise) under the alternative.

Next the critical RFT threshold can be computed using **power1d**'s inverse survival function following *Friston et al. (1996)* (Eqn. 5, p. 226) as follows:

```
u = power1d.prob.t_isf( alpha , df , Q , W0 )
```

Next the smoothness and threshold parameters are transformed according to *Friston et al. (1996)* (Eqns. 8–9, p. 227):

```
s2 = sigma
f = float( W1 ) / W0
Wstar = W0 * ( ( 1 + s2 ) / (1 + s2 / ( 1 + f ** 2 ) ) ) ** 0.5
ustar = u * ( 1 + s2 ) ** -0.5
```

Here *s2* is the variance and *f* is the ratio of signal-to-noise smoothness. The probability of rejecting the null hypothesis when the alternative is true is given as the probability that random fields with smoothness $W^*$ will exceed the threshold $u^*$ (*Wstar* and *ustar*, respectively), and where that probability can be computed using the standard RFT survival function:

```
p = power1d.prob.t_sf( ustar , df , Q , Wstar )
```

Here the analytical power is $p = 0.485$. Validating this analytical power calculation in **power1d** can be achieved using a null signal and two different noise models as follows:

```
baseline = power1d.geom.Null( Q=Q )
signal = power1d.geom.Null( Q=Q )

SG = power1d.noise.SmoothGaussian
n0 = SG( Q=Q , sigma=1.0 , fwhm=W0 , J=J )
n1 = SG( Q=Q , sigma=1.0 , fwhm=Wstar , J=J )

model0 = power1d.models.DataSample( baseline , signal , n0 , J=J )
model1 = power1d.models.DataSample( baseline , signal , n1 , J=J )

teststat = power1d.stats.t_1sample
emodel0 = power1d.Experiment( model0 , teststat )
emodel1 = power1d.Experiment( model1 , teststat )

sim = power1d.ExperimentSimulator( emodel0 , emodel1 )
results = sim.simulate( 1000 )
p = results.sf( ustar )
```

The numerically estimate power is $p = 0.492$, which is reasonably close to the analytical probability of 0.485 after just 1,000 iterations. Repeating for background noise smoothness values of 10, 20 and 50, sample sizes of 5, 10 and 25 and effect sizes ranging from $\sigma = 0.5$ to 2.0 yields the results depicted in Fig. 15. Close agreement between the theoretical and simulated power results is apparent. As noted by *Hayasaka et al. (2007)* powers are quite low for the inflated variance approach because the signal is not strong; the 'signal' is effectively just a different type of noise. The noncentral RFT approach described in the next section addresses this limitation.

## 1D power: noncentral RFT method

The noncentral RFT method models signal as a constant continuum shift (*Hayasaka et al., 2007*). Like the inflated variance method above, it can by computed analytically in **power1d** by first defining all power-relevant parameters:

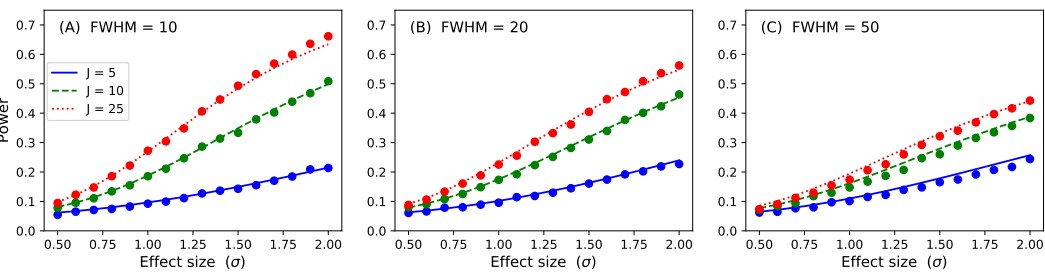

**Figure 15  Validation results for the inflated variance approach to 1D power.** Solid lines depict theoretical solutions from the noncentral random field theory and dots depict **power1d**'s numerically simulated results (10,000 iterations each). *J* represents sample size and FWHM represents the smoothness of the background noise process. (A) FWHM = 10, (B) FWHM = 20, (C) FWHM = 50.

```
J = 8
Q = 201
W = 40.0
df = J - 1
alpha = 0.05
effect = 0.8
delta = effect * J ** 0.5
```

where *delta* is the noncentrality parameter. Next power can be be computed via noncentral RFT (*Hayasaka et al., 2007*; *Mumford & Nichols, 2008*; *Joyce & Hayasaka, 2012*) as follows:

```
u = power1d.prob.t_isf( alpha , df , Q , W )
p = power1d.prob.nct_sf( zstar , df , Q , W , delta )
```

Here *u* is the critical threshold and *nct_sf* is RFT's noncentral *t* survival function. The analytical power is $p = 0.731$. Next, similar to the 0D validation above, **power1d** can be used to validate this analytical power by constructing signal and noise objects as indicated below. Note that the signal is *Constant* (Fig. 5), as assumed by the noncentral RFT method.

```
baseline = power1d.geom.Null( Q )
signal0 = power1d.geom.Null( Q )
signal1 = power1d.geom.Constant( Q , amp=effect )
n = power1d.noise.SmoothGaussian( J , Q , mu=0 , sigma=1 , fwhm=W )
model0 = power1d.DataSample( baseline , signal0 , n , J=J )
model1 = power1d.DataSample( baseline , signal1 , n , J=J )
```

Last, simulate the modeled experiments and numerically estimate power:

```
teststat = power1d.stats.t_1sample
emodel0 = power1d.models.Experiment( model0 , teststat )
emodel1 = power1d.models.Experiment( model1 , teststat )
sim = power1d.ExperimentSimulator( emodel0 , emodel1 )
```

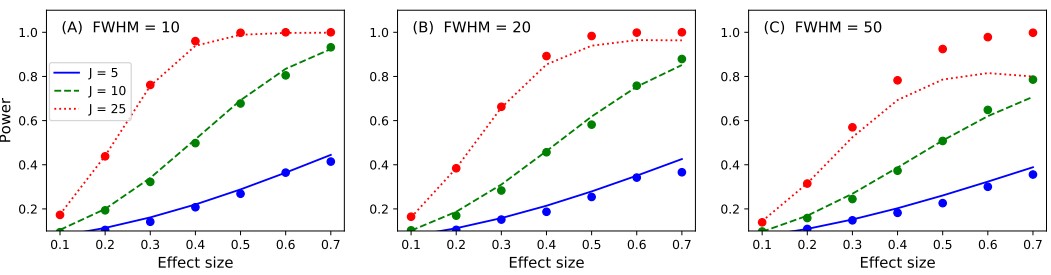

**Figure 16** **Validation results for the noncentral random field theory approach to 1D power.** Solid lines depict theoretical solutions from the noncentral random field theory and dots depict **power1d**'s numerically simulated results (10,000 iterations each). FWHM and $J$ represent continuum smoothness and sample size, respectively. (A) FWHM = 10, (B) FWHM = 20, (C) FWHM = 50.

```
results = sim.simulate( 1000 )
p = results.p_reject1
```

Here the numerically estimated power is $p = 0.747$, which is again similar to the analytical probability of $p = 0.731$ after just 1,000 iterations. Repeating for smoothness values of 10, 20 and 50, sample sizes of 5, 10 and 25 and effect sizes ranging from 0.1 to 0.7 yields the results depicted in Fig. 16. Agreement between the theoretical and numerically simulated powers is reasonable except for large effect sizes and intermediate sample sizes (Fig. 16C, $J = 25$). Since theoretical and simulated results appear to diverge predominantly for high powers these results suggest that the noncentral RFT approach is valid in scenarios where powers of approximately 0.8 are sought for relatively small sample sizes.

While the noncentral RFT approach has addressed the low-power limitation of the inflated variance method (Fig. 15), its 'signal' is geometrically simple in the form of a mean shift. Clearly other, more complex signal geometries may be desirable. For example, in the context of the Canadian temperature data (Fig. 1) one may have a forward dynamic model which predicts regional temperatures through region-specific parameters such as land formations, foliage, wind patterns, proximity to large bodies of water and atmospheric carbon dioxide. Forward models like those can be used to generate specific continuum predictions based on, for example, increases in atmospheric carbon dioxide. Those continuum predictions are almost certainly not simple signals like the ones represented by the inflated variance and noncentral RFT methods. Therefore, when planning an experiment to test continuum-level predictions, and specifically when determining how many continuum measurements are needed to achieve a threshold power, the numerical simulation capabilities of **power1d** may be valuable.

## COMPARISON WITH OTHER SOFTWARE PACKAGES

Power calculations for 0D (scalar) data are available in most commercial and open-source statistical software packages. Many of those offer limited functionality in that most are limited to the noncentral $t$ distribution, and many have vague user interfaces in terms of experimental design. Some also offer an interface to noncentral $F$ computations, but nearly all have limited capabilities in terms of design.

The most comprehensive and user-friendly software package for computing power is G-power (*Faul et al., 2007*). In addition to the standard offerings of noncentral $t$ computations, G-power also offers noncentral distributions for $F$, $\chi^2$ and a variety of other test statistics. It has an intuitive graphical user interface that is dedicated to power-specific questions. However, in the context of this paper G-power is identical to common software packages in that its power calculations are limited to 0D (scalar) data.

Two software packages dedicated to continuum-level power assessments, and those most closely related to power1d are:

1. PowerMap (*Joyce & Hayasaka, 2012*).
2. fmripower (*Mumford & Nichols, 2008*).

Both PowerMap and fmripower are designed specifically for continuum-level power analysis, and both extend the standard noncentral t and F distributions to the continuum domain via RFT. They have been used widely in the field of Neuroimaging for planning brain imaging experiments and they both offer graphical interfaces with a convenient means of incorporating piiot data into guided power analyses. However, both are limited in terms of the modeled signals they offer. RFT's noncentral t and F distributions model 'signal' as a whole-continuum mean displacement, which is geometrically simple relative to the types of geometries that are possible at the continuum level (see the 'Software Implementation: Geometry section above). PowerMap and fmripower somewhat overcome the signal simplicity problem through continuum region constraints, where signal is modeled in some regions and not in others in a binary sense. This approach is computationally efficient but is still geometrically relatively simple. A second limitation of both packages is that they do not support numerical simulation of random continua. This is understandable because it is computationally infeasible to routinely simulate millions or even thousands of the large-volume 3D and 4D random continua that are the target of those packages' power assessments. Consequently neither PowerMap nor fmripower supports arbitrary continuum signal modeling.

As outlined in the examples above **power1d** replicates the core functionality of PowerMap and fmripower for 1D continua. It also offers functionality that does not yet exist in any other package: arbitrary continuum-level signal and noise modeling and associated computational power analysis though numerical simulation of random continua. This functionality greatly increases the flexibility with which one can model one's data, and allows investigators to think about the signal and noise in real-world units, without directly thinking about effect sizes and effect continua.

## SUMMARY

This paper has described a Python package called **power1d** for estimating power in experiments involving 1D continuum data. Its two main features include (a) analytical continuum-level power calculations based on random field theory (RFT) and (b) computational power analysis via continuum-level signal and noise modeling. Numerical simulation is useful for 1D power analysis because 1D continuum signals can adopt arbitrary and non-parameterizable geometries. This study's cross-validation results show

that **power1d**'s numerical estimates closely follow theoretical solutions, and also that its computational demands are not excessive, with even relatively complex model simulations completing in under 20 s. Since **power1d** accommodates arbitrary signals, arbitrary noise models and arbitrarily complex experimental designs it may be a viable choice for routine yet flexible power assessments prior to 1D continuum experimentation.

### Funding
This work was supported by Wakate A Grant 15H05360 from the Japan Society for the Promotion of Science. The funders had no role in study design, data collection and analysis, decision to publish, or preparation of the manuscript.

### Grant Disclosures
The following grant information was disclosed by the author:
Japan Society for the Promotion of Science: 15H05360.

### Competing Interests
The author declares there are no competing interests.

### Author Contributions
- Todd C. Pataky conceived and designed the experiments, performed the experiments, analyzed the data, contributed reagents/materials/analysis tools, wrote the paper, prepared figures and/or tables, performed the computation work, reviewed drafts of the paper, wrote the software and its documentation.

### Data Availability
All code, including scripts to replicate the paper's results, are available at http://www.spm1d.org/power1d.

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
