# Peer review of "Power1D: a Python toolbox for numerical power estimates in experiments involving one-dimensional continua"

_PeerJ Computer Science, doi:10.7717/peerj-cs.125_

## Round 0.1 · original submission · Minor Revisions

· Academic Editor

Minor Revisions

The external peer reviewers evaluated your manuscript very positively, and I agree. Please respond timely to the comments, I look forward to quickly process your revised paper.

·

Basic reporting

The manuscript submitted introduces a 1D power estimation toolbox. It is clearly written and will be relevant for future studies within many fields.

Figures are clear and support the understanding of the paper, while not all y axis are labeled (4, 5, 6, 7, 8, 9,10, 12, 13) and different line styles would help the understanding of graphs when printed black-white.

As I am not as experienced with the mathematics that underpin the ideas behind code, I would have appreciated a bit more detail regarding what findings means - e.g. what values are we looking for?

Experimental design

The research question is well defined and the conclusion answers/addresses the purpose of the paper.

Methods applied in the paper are well described and proven using simulations. Code is made available allowing the paper to be replicated.

Validity of the findings

The processes introduced within the paper are sound and provide the user a ability to easily perform a 1d power estimation of a waveform.

Conclusion are well stated and differences to existing platforms are explained and justify benefit of the introduced methods and hence the need for publication.

Additional comments

Congratulations Todd to a well written and interesting paper.

Below you'll find minor suggestion that will eliminate a few typos and might make the paper a bit easier to read for users from a sport science background.

Could you provide a code with would provide an answer to the number of samples needed for an X% increase of a peak value?

Line 49: consider "q represents the day of the ..."
Line 66: "analytical COMMA one can ...."
Line 69-71: I am not sure if I got this right but there is a comma missing: "behavior COMMA ....."
Line 84: remove "also"
Line 84 and following: you use absolute terms use incorrect rather then clearly incorrect, while incorrect should be replace with inappropriate. This should be adapted in the following lines
Line 107: do you mean "model a signal as ...." ? The same in line 109.
Line 208 and following please review the style of the term DataSample and make it consistent.
Line 301: You might consider to mention / explain Monte Carlo simulation and the Omnibus test before mentioning it - as your techniques will be used in a sport science world.

Figure 12: p = 0.992 is not correct. Please explain null and alternative in the text before using it in the figure. Why did you choose COI = 3? What are the dotted line in the figure?

Line 527: "(see ยง)" should refer to something
Line 530: consider to omit "understandable"

·

Basic reporting

In the submitted manuscript, "Power1D: A Python toolbox for numerical power 1 estimates in experiments involving one-dimensional continua", the author presents a computer package called "power1d", which implements analytical 1D power solutions and develops a computational approach to continuum-level power analysis, which allows arbitrary signal and noise modeling.

Use of English is correct, clear and unambiguous. The introduction gives a good account of the statistical and computational problems that arise in continuum measurements. The need for the computational methods implemented in the package is well motivated with real data clearly explained. Figures are well designed, relevant and informative. Literature is relevant and well referenced.

Raw data as well as source code are supplied and well documented into the site of the package.

MINOR POINT:

End of line 45, continuing in line 46, substitute "to not vary completely independently" by "do not vary in a completely independent way".

Experimental design

The author proposes, implement, test and demonstrates a computational approach to continuum-level power analysis that permits arbitrary signal and noise modeling, which is clearly within the scope of "PeerJ Computer Science".

Implementation, dependencies and capabilities of the package are well described in the manuscript and more details are given in the site of the package (http://www.spm1d.org/power1d). The research question is well defined and its relevance is clear, while the solutions implemented in the package are well supported. The research presented fills a gap in the methods for continuum-level power analysis, as the author shows both, in the introduction as well as in the discussion of the results.

The author comprehensively demonstrates the use of the package for distinct scenarios, clearly showing the frameworks where the methods are of application. The use of real data makes the presentation easier to understand for statisticians and other interested in this topic. The use of a random seed makes the results replicable; in my opinion the research was carried out in a rigorous way with very good technical standards. The methods are described in detail, and the availability of the source code makes it possible to evaluate all details of the methods and their implementation.

Validity of the findings

Validations of the methods implemented in "Power1D" are clearly and extensively shown for 0D power, the 1D power with inflated variance and non-central RFT. Comparisons with other software packages are complete and fair; to my knowledge the author surveyed the main alternative packages, stating their limitations, some of which are filled by the methods implemented in "Power1D". In particular, the implementation of arbitrary continuum-level signal and noise modeling, which allows the power analysis though numerical simulation of random continua, is a remarkable characteristic of the described package.

The methods presented by the author and implemented in the package are statistically sound and well presented and exemplified by the use of both, simulations as well as real data. Conclusions are well stated, linked to the original research question and limited to supporting results.

---

## Round 0.2 · accepted · Accept

· Academic Editor

Accept

I will also test Power1D for the analysis of biological mass spectrometry data (at-line monitoring of natural compounds) in my group.